# Eyes Wide Open: Assessing Early Visual Behavior in Zebrafish Larvae

**DOI:** 10.3390/biology14080934

**Published:** 2025-07-24

**Authors:** Michela Giacich, Maria Marchese, Devid Damiani, Filippo Maria Santorelli, Valentina Naef

**Affiliations:** Neurobiology and Molecular Medicine Unit, IRCCS Fondazione Stella Maris, Calambrone, 56128 Pisa, Italy; michela.giacich@fsm.unipi.it (M.G.); maria.marchese@fsm.unipi.it (M.M.); devid.damiani@fsm.unipi.it (D.D.)

**Keywords:** *Danio rerio*, neurodevelopment, neurodegeneration, retina, behavior

## Abstract

Neurodegenerative diseases affect the brain and nervous system, frequently causing vision impairment at early stages. Detecting these conditions promptly is important because it allows for better treatment and an improved quality of life. The retina, a part of the eye connected to the brain, can show signs of these diseases and can be studied without invasive methods. Zebrafish, a small fish with transparent embryos, have become an important model for studying eye and brain disorders. Their visual system develops quickly and can be easily observed through behavioral tests that measure how larvae respond to light and colors. This review focuses on behavioral tests performed on zebrafish larvae throughout the early phases of embryonic and larval development. These tests provide sensitive and rapid methods to identify vision problems before clear damage occurs. However, differences in how these tests are performed across laboratories highlight the need to standardize methods for reliable results. Combining behavioral observations with imaging and electrical recordings will improve our understanding of vision and help develop new treatments. This review summarizes the main behavioral assays used in zebrafish larvae to study eye function and highlights their potential for early diagnosis and drug screening in neurodegenerative diseases.

## 1. Introduction

Retinal degeneration is a pathological process affecting the retina, sometimes preceding cognitive or motor symptoms in diverse neurodegenerative diseases [1,2]. Conditions such as juvenile Parkinson’s disease, cerebellar ataxias and spastic paraplegias exhibit retinal involvement that can be detected long before widespread neuronal damage occurs [2]. Moreover, characteristic retinal changes are closely associated with cognitive decline in disorders such as Alzheimer’s disease, Huntington’s disease, and Joubert syndrome [3]. In addition to these, several other neurodegenerative diseases present with visual alterations, often reflecting underlying retinal or optic nerve involvement. The most relevant of these conditions are summarized in Table 1.

Because the retina is an accessible part of the central nervous system (CNS), it serves as a valuable platform for studying neurodegenerative disorders. Neurological diseases often manifest ocular symptoms due to degeneration of visual pathways, involving photoreceptors, retinal ganglion cells, and other retinal neurons [36]. Brain neurodegenerative processes—such as oxidative stress, neuroinflammation, protein aggregation, and mitochondrial dysfunction—are also reflected in retinal pathology [37].

Retinal alterations—including ganglion cell loss, thinning of the retinal nerve fiber layer, and vascular changes—have been proposed as biomarkers for neurodegenerative diseases, enabling earlier diagnosis and intervention [38]. Compared to the brain, the retina is more accessible, offering a practical method to monitor disease progression and support the development of targeted therapies [2].Among vertebrate models, the zebrafish (*Danio rerio*) has emerged as a key model for studying retinal degeneration, providing unique opportunities for investigating disease mechanisms and drug discovery [39,40]. Moreover, as the retinal development in zebrafish reflects that of humans, impairments occurring in neurodevelopmental diseases can also be evaluated. Examples include abnormal blood vessel growth as in retinopathy of prematurity (ROP); the thickening of all retinal layers as in patients affected by autism spectrum disorder (ASD); and global developmental delay, as in neurodevelopmental disorders with eye movement abnormalities and ataxia (NEDEMA) [41,42,43]. Despite their smaller eyes, zebrafish possess nearly all the fundamental structures found in human eyes, sharing the same layered architecture. Both zebrafish and human retinas consist of five distinct layers: three cellular layers—the outer nuclear layer (ONL), inner nuclear layer (INL), and ganglion cell layer (GCL)—and two synaptic layers—the inner and outer plexiform layers (IPL and OPL) [44]. Additionally, the zebrafish retina contains the same major classes of retinal neurons as humans, including retinal ganglion cells (RGCs), bipolar cells, horizontal cells, and amacrine cells, as well as essential glial elements such as Müller cells, astrocytes, and microglia [39]. The zebrafish retina is densely packed with cone photoreceptors that reflect human macula, supporting high-acuity and color vision. It contains four distinct cone types: blue (SWS2), ultraviolet (SWS1), green (RH2), and red (LWS), with green and red cones forming physically fused double cones [44]. The zebrafish visual system matures rapidly, becoming fully functional by 72 h post-fertilization (hpf), when retinal morphology and function closely resemble those of adult humans [45]. Visual information from photoreceptors is processed through the retinal layers, reaching the ganglion cells before being transmitted to the brain [46]. The structural similarity between zebrafish and human retinas, combined with the practical advantages of the zebrafish model, highlights its utility in studying visual behavior, neural circuitry, and disease pathogenesis [40,47]. As a result, zebrafish provide a powerful tool for exploring the intricate relationship between the retina and other CNS regions, offering insights that may lead to new therapeutic strategies for neurodegenerative disorders. By summarizing recent advancements and methodologies, this narrative review will highlight the significance of behavioral assays for assessing retinal function in zebrafish during early developmental stages, focusing on their pivotal role in screening putative therapeutic strategies to tackle neurodegenerative processes. Integrating diverse approaches, this review seeks to inspire new research directions and innovative experimental designs in vision science and neurodegeneration.

## 2. Mechanisms of Retinal Degeneration in Neurodegenerative Diseases

Retinal manifestations of structural and functional degeneration are closely linked to the progression of neurodegenerative disorders [48]. Both the retina and optic nerve develop as paired evaginations from the anterior central nervous system during embryogenesis [49]. As a result, the retina and the brain share a common cellular origin, making the retina an integral part of the CNS. Retinal and brain neurons share many structural and functional similarities. For instance, retinal neurons depend on intricate synaptic networks for communication [50], utilizing various neurotransmitters such as acetylcholine, dopamine, glutamate, glycine, and gamma-aminobutyric acid (GABA) [51]. The retina processes visual stimuli by converting light signals into electrical impulses, which are then transmitted to the brain for further interpretation. This process mirrors how neurons in the brain receive, process, and transmit different types of sensory information. The last part of retinal processing occurs in RGCs, which form complex dendritic networks in the IPL and convey the signal to thalamic neurons to relay visual information [52]. Beyond this primary function, retinal neurons also contribute to modulating reflexive and adaptive behaviors, reflecting the role of brain neurons in controlling movement, cognition, and sensory perception. Moreover, retinal and cerebral blood vessels exhibit notable similarities in several key areas, including the angles and organization of capillary branching [53]. Given these similarities, the mechanisms that occur in retinal degeneration closely mirror those observed in neurodegenerative diseases of the CNS. One of the most prominent mechanisms is oxidative stress, which plays a central role in both retinal and brain degeneration. The retina is highly metabolically active and, as such, is particularly vulnerable to damage from reactive oxygen species (ROS) [54]. An increase in ROS production can lead to cellular damage, inflammation, and, ultimately, retinal neuron death, a process that reflects what happens in the brain [55]. Another shared mechanism is protein aggregation, a hallmark of conditions such as Retinitis pigmentosa (RP), Alzheimer’s disease, and Parkinson’s disease [56]. Neuroinflammation is also a critical factor common in both the brain and retina, where the activation of microglia and retinal macrophages triggers inflammation that worsens neuronal damage [57]. In particular, chronic inflammation in the retina can impair the blood-retina barrier (BRB), leading to vascular changes and further degeneration of retinal tissue [58]. This process reflects similar inflammatory mechanisms seen in neurodegenerative diseases of the CNS. Moreover, axonal degeneration is a feature shared by both retinal and brain neurodegeneration. In the brain, axonal damage is a hallmark of multiple diseases, including multiple sclerosis, while in the retina, degeneration of RGCs axons is an early event, characterizing common conditions such as glaucoma [59,60]. These shared mechanisms—oxidative stress, protein aggregation, neuroinflammation, and axonal degeneration—highlight the strong functional and pathophysiological connections between the retina and the brain (Figure 1). In this context, behavioral assays in zebrafish could offer a rapid, non-invasive method to assess visual function at early developmental stages, enabling early detection of retinal dysfunction. These tests contribute to a better comprehension of neurodegenerative processes and serve as invaluable tools for high-throughput screening of potential therapeutic compounds aimed at halting or reversing retinal degeneration.

## 3. Zebrafish Retinal Development Overview

Retinal development in zebrafish begins during gastrulation when the optic vesicles evaginate from the forebrain. By 12 hpf, these vesicles undergo morphogenetic movements to form the optic cup, which subsequently differentiates into the neural retina and retinal pigment epithelium by 24 hpf [61,62]. The retinal neuroepithelium then proliferates rapidly, laying the foundation for developing distinct retinal layers. Neurogenesis in the zebrafish retina follows a conserved pattern observed in vertebrates, beginning at 28 hpf. The first cells to differentiate are the RGCs by 32 hpf at the level of the ventro-nasal part of the zebrafish retina, and establish connections to the optic tectum [63]. Neurogenesis continues from the ventro-nasal part through the rest of the retina and by 60 hpf, the distinct layersouter photoreceptor, inner nuclear, and ganglion cell layersbecome evident. These fundamental layers are separated by two layers of synapses, the inner and outer plexiform layers [64]. Zebrafish retinal differentiation is nearly completed by 72 hpf, with all principal neuronal and glial cell types present [65]. Between 72 and 120 hpf, the zebrafish retina undergoes significant functional maturation. Synaptic connections are established within the inner and outer plexiform layers, and photoreceptor outer segments begin to develop, allowing zebrafish larvae to respond to light stimuli. Thus, at 120 hpf the visual synaptic system is fully functional [44]. Together, these morphologic developments and synaptic connections allow for the integration of visual signals and their relay to specific brain regions, facilitating complex visual behaviors and reflexes [64]. Unlike nocturnal rodents such as mice and rats, which have retinas with relatively few cones, cone photoreceptors dominate the zebrafish retina, as in humans, making it an excellent model for studying cone-mediated vision and macular disorders [66]. Zebrafish possess four types of cones (blue, UV, and red/green double cones) and one rod cell type [67]. Notably, the additional type of cone cell sensitive to UV light, absent in humans, enhances their visual capabilities in aquatic environments and is specifically tuned for prey capture [68]. Visual signals originating in the photoreceptors layer are transmitted through the retina to the ganglion cells, processing the refined visual information and transmitting it to the brain via the optic nerve [63]. Several key brain regions, including the optic tectum, act as a key visuomotor hub [69]. The tectum is characterized by nine input layers that process data for complex spatial and motion-related responses, setting up behaviors like prey capture and predator avoidance [69]. The pretectal area is primarily associated with the visual system and plays a key role in visuomotor behaviors. It receives inputs from the retina and the optic tectum and integrates information related to visual stimuli. This system coordinates visual inputs with motor responses, such as the optokinetic (OK) and optomotor (OM) responses. Indeed, after processing visual information, the pretectal area sends signals to motor neurons located in the oculomotor nuclei and in the nucleus of the medial longitudinal fasciculus (nMLF), which then produce appropriate motor reactions [70]. The nMLF is involved in coordinating swimming and escape responses. The nucleus isthmi provides cholinergic feedback to the optic tectum and pretectum, supporting stimulus selection and persistence during prey capture and selective attention tasks (Figure 2). Furthermore, specific regions, such as the preoptic area of the hypothalamus, play significant roles in modulating behaviors with visual signals, contributing to hormonal control involved with circadian rhythms [69,71]. Together, these diversified layers provide a highly organized and efficient system for visual processing, shaping the fish behavioral responses [40,47,72].

This well-characterized timeline of retinal development provides the foundation for implementing standardized, low-cost, and easily reproducible behavioral assays that serve as rapid and sensitive readouts of retinal function at defined developmental stages (Figure 3). These assays span from early light responsiveness to more complex visuomotor behaviors, offering a valuable tool set for systematically probing visual system integrity in zebrafish. Each of these behavioral paradigms will be described in the following sections.

## 4. Behavioral Assays to Study Retinal Function in Zebrafish from 24 h Post-Fertilization to the Juvenile Stage

One of the key challenges in neurodegenerative diseases of the central nervous system is their early detection and accurate diagnosis. Although these disorders affect various functional systems, many also involve impairments in oculomotor function, due to the vulnerability of the oculomotor system to toxic protein aggregates and other pathophysiological mechanisms underlying neurodegeneration [1,49,73]. As a result, assessing oculomotor performance has become an important tool for diagnosing several neurodegenerative conditions and monitoring disease progression [73]. The zebrafish oculomotor system is highly conserved and shares fundamental mechanisms with that of humans. Its robust responsiveness to visual stimuli enables precise evaluation of oculomotor deficits associated with neurodegenerative processes [74,75]. Zebrafish depend heavily on vision for essential behaviors such as prey capture, predator avoidance, schooling, and responses to motion and ambient light. Several behavioral assays now allow for non-invasive assessment of visual function across different developmental stages. These tests yield real-time data on visual responses and are instrumental in identifying early functional impairments. While research in this area was historically constrained by limited experimental tools and techniques, recent technological advances have substantially accelerated progress. These innovations have provided researchers with powerful and user-friendly platforms for more effective behavioral testing. In the following sections, we summarize some of these tools (Table 2), which are specifically designed to serve as convenient references for evaluating retinal function during the early stages of zebrafish development.

### 4.1. Visual Background Adaptation (VBA) Assay

The Visual background adaptation test (VBA) evaluates the zebrafish’s pigmentary responses to varying light conditions by examining melanosome dispersion or aggregation within melanophores, cells responsible for skin and eye pigmentation [76]. This assay can be carried out by placing the larvae in Petri dishes or multi-well plates against bright or dark backgrounds, allowing melanosomes to adapt to environmental light. Following an adaptation period of 15–30 min, researchers can measure pigmentary responses using imaging systems, such as confocal or brightfield microscopes equipped with digital imaging capabilities, and software packages like FIJI or specialized pigmentation quantification tools for quantitative analysis. Indeed, the clear visibility of melanocytes in zebrafish larvae makes them ideal for studying these processes through high-resolution imaging analyses. Studies have shown that these pigmentary responses are critical for camouflaging and protecting against UV radiation. Under bright light conditions, retinal ganglion cells (RGCs) signal the hypothalamus to release MCH, which causes melanosomes to aggregate near the nucleus, lightening the fish’s appearance. Conversely, in dark environments, RGCs signal the release of corticotropin-releasing factor (CRF), which stimulates the anterior pituitary to secrete α-MSH. This hormone promotes the dispersion of melanosomes throughout the cytoplasm, darkening the fish’s skin and enhancing survival by reducing predation risk [77,78,79]. This response can be modulated through environmental changes and pharmacological interventions. For instance, ethanol has been shown to override light-induced signals and activate the CRF pathway, leading to melanosome dispersion even under bright conditions. These pathways can be further explored using agonists, antagonists, or gene knockdown approaches to disrupt MCH or α-MSH signaling. For example, antalarmin and K414198—CRF antagonists—significantly reduce dark-induced melanosome dispersal, confirming CRF’s essential role in regulating camouflage behavior [77]. Furthermore, hypopigmented mutants lacking melanophores or iridophores exhibit impaired visual sensitivity under extreme lighting conditions, emphasizing the role of retinal and skin pigments in modulating visual responses [80]. Environmental stressors, such as pollutants or changes in UV exposure, can also impair melanosome dynamics, affecting both camouflage and stress responses. These effects highlight the importance of adaptation mechanisms in evolutionary processes across diverse aquatic environments. Overall, these pigmentary changes provide an indirect measure of photoreceptive function and neural control over light adaptation mechanisms. By analyzing melanosome dynamics, the VBA assay reveals the intricate interactions between environmental light, neuroendocrine signaling, and pigmentary adaptation.

### 4.2. Prey Tracking and Capture Assay

In zebrafish larvae, prey capture is an innate behavior that emerges around 4 days post-fertilization (dpf), coinciding with the onset of swimming. Despite its apparent simplicity, this behavior involves complex neural processes, including visual perception, recognition, decision-making, and motor control [81,82]. Zebrafish larvae exhibit a highly stereotyped response when interacting with prey, such as paramecia. Upon detecting prey, the fish reorients its body through a series of unilateral tail flicks (J-bends) and forward swims, gradually moving toward a proximal striking zone. The final action involves a rapid dart forward to engulf the prey in a capture swim [83]. Prey capture behavior in zebrafish larvae can be experimentally assessed using UV light stimuli. Larvae are typically immobilized in agarose, allowing free movement of the eyes and mouth. This setup enables the controlled presentation of a UV stimulus that mimics prey movement and reliably elicits hunting and strike responses [84]. The experimental configuration includes a behavior chamber, a Light Crafter projector for precise UV stimulus delivery, and synchronized cameras to capture key behavioral metrics such as eye convergence, head elevation, and strike initiation [85]. Vision plays a primary role in guiding this behavior. Eye convergence increases as the fish approaches its target, enhancing depth perception and spatial precision [86]. This expansion of binocular visual-field coverage is crucial for successful targeting and capture. Studies using virtual prey (e.g., moving dots) have shown that even restrained larvae exhibit eye convergence and tail movements, indicating that visual input alone is sufficient to trigger the full prey capture sequence [87,88]. Visual information about prey location is transmitted from the retina to two major contralateral visual areas: the pretectum and the optic tectum (OT) [89]. Disruption of this pathway significantly impairs prey capture. For instance, ablation of retinal input to the OT markedly reduces prey capture success. Similarly, blumenkohl (blu) mutant larvae, which harbor a mutation in a gene encoding a vesicular glutamate transporter expressed in retinal ganglion cells, show impaired prey capture, especially with small prey like paramecia [90]. These mutants also exhibit reduced sensitivity to high spatial and temporal frequencies and display abnormal retinotectal arborization, including enlarged axonal arbors and broader receptive fields, leading to a coarser retinotopic map and impaired visual processing [90]. Altogether, these findings highlight the central role of visual processing in mediating zebrafish prey capture behavior and underscore the utility of this behavioral assay in assessing visual system function and its developmental or genetic perturbations.

### 4.3. Escape Response (ER) Assay

The escape response (ER) is an innate, highly stereotyped behavior that allows animals to evade potential threats. In zebrafish, it is triggered by sudden visual stimuli and depends on precise coordination between sensory input and motor output. Because it demands both speed and adaptability, the ER has become a powerful model for studying how visual information is processed and converted into rapid behavioral responses. This test involves placing larvae inside a circular rotating drum featuring a single black stripe that mimics an approaching threat [91]. Zebrafish instinctively attempt to avoid the stripe, and their movement patterns vary depending on the size and dynamics of the stimulus. This setup provides a controlled environment to assess visual threat perception and sensorimotor integration. Zebrafish larvae are particularly well-suited for investigating the neural basis of the ER. Their optical transparency allows real-time imaging of neuronal activity, offering insights into how visual signals are processed by the brain to produce escape behaviors. At the core of the ER lies a well-defined neural circuit involving the retina, thalamus, optic tectum, and hindbrain, which together mediate the detection of visual threats and the execution of escape maneuvers [92]. Various experimental approaches have been developed to investigate this response in zebrafish, including interactions with real predators, virtual simulations displayed on screens, and robotic predator models [92]. A widely used method involves placing zebrafish inside a circular rotating drum featuring a single black stripe that mimics an approaching threat [91]. The fish instinctively attempt to avoid the dark stripe, with their movement patterns varying depending on the size and dynamics of the stimulus. Beyond threat detection, the ER assay has also been instrumental in identifying genetic mutations that impair visual function. For example, it has helped characterize dominant mutations like nba (night blindness a) and nbb (night blindness b), both of which are associated with progressive retinal degeneration [93]. In addition, this method has revealed how the circadian clock influences visual sensitivity. Studies measuring the minimum light intensity required to trigger an escape response across a 24-h cycle found that zebrafish are most responsive to visual stimuli in the late afternoon—just before nightfall—and least sensitive at dawn, prior to the transition to daylight [94]. These findings underscore the role of circadian rhythms in modulating visual processing and behavior.

### 4.4. Fundamental Visual Reflexes: The OKR and OMR

Abnormalities in optokinetic nystagmus (OKN) are a well-documented clinical finding in various neurological disorders, often reflecting damage to retinal cells or neural pathways connecting the eye to the brain [95]. In neurodegenerative conditions such as Alzheimer’s disease, mild cognitive impairment, Parkinson’s disease, amyotrophic lateral sclerosis, frontotemporal dementia, and vascular cognitive impairment, these alterations can manifest as impaired smooth pursuit, reduced eye movement coordination, or even complete loss of response [73]. In this assay, zebrafish larvae are immobilized to restrict body movement while allowing free eye motion [65,96]. The response is elicited by presenting a moving visual stimulus, typically rotating black-and-white stripes that span a large portion of the visual field. Vertically oriented gratings evoke a horizontal OKR, while horizontally oriented gratings trigger a vertical OKR. The response includes two distinct phases: a slow phase, during which the eyes smoothly follow the moving stripes to minimize retinal slip, and a fast phase (saccade), which rapidly repositions the eyes in the opposite direction [74,97]. This experimental setup allows precise modulation of visual parameters such as spatial frequency, contrast, and angular velocity, enabling detailed assessment of visual function. Several studies have demonstrated that zebrafish larvae reliably exhibit the optokinetic response (OKR) as early as 72 hpf, a developmental stage at which retinal circuits and the optic tectum have reached functional maturity [47,98,99,100]. By 96 hpf, the OKR becomes more robust, with improved stimulus tracking duration and increased eye movement velocity [63]. Despite its widespread use, standardized reporting guidelines for OKR assays have yet to be fully established. Recently, Rodwell and colleagues performed a systematic literature review to identify optimal reporting standards, promoting reproducibility and transparency in OKR experimental methods [101]. In addition to OKR, the optomotor response (OMR) serves as another key behavioral assay for evaluating visual function.

The OMR is an innate reflex observed in numerous species, allowing animals to adjust their movement in response to large-scale visual motion, thereby maintaining spatial stability [102]. Unlike OKR, which measures eye movements, OMR assesses locomotor behavior as the primary response. This reflex involves binocular neural activation and is mediated by red and green cone photoreceptors, with processing occurring in midbrain visual centers [103,104]. OMR assays have become widely used in preclinical models to evaluate visual and motor function in neurological disorders [105]. Zebrafish larvae exhibit OMR between 5 and 7 dpf [106], with responses becoming more robust at later stages, particularly at 10 and 13 dpf, compared to stationary background conditions [107]. When placed in an arena with moving patterns, zebrafish instinctively align their swimming direction with the motion to stabilize their visual field [108]. This response is highly relevant for analyzing impairments in visual-motor coupling caused by genetic mutations or pharmacological treatments [40,109]. OMR assays require careful calibration of spatial frequency and angular velocity based on the developmental stage of zebrafish. For early-stage larvae (24–48 hpf), lower speeds are preferable due to the immaturity of the visual and motor systems. Conversely, at later stages (120 hpf), higher speeds can be used to assess fully developed motion tracking and visual-motor integration. Studies have determined that the optimal setting for 120 hpf larvae is 16 angular cycles at 1.04 rad/s [107]. Regarding visual acuity and motion detection, different stimulus speeds can be used to test specific functional aspects. Low-speed stimuli help identify vision-impaired mutants at early developmental stages and determine minimal motion perception thresholds. In contrast, high-speed stimuli assess advanced motion tracking abilities, including directional following, speed sensitivity (where zebrafish adjust their swimming speed proportionally to the perceived motion), and spatial pattern recognition using stripes of varying widths to evaluate visual acuity. Additional challenges, such as high-speed or low-contrast patterns, allow researchers to explore more complex visual processing and motor responses. To ensure accuracy in OMR measurements, zebrafish responses should be recorded for 10–20 min using high-resolution cameras positioned above the experimental setup. Optimizing arena construction can further enhance experimental precision: anti-glare screens help minimize reflections, while careful contrast calibration ensures consistent stimulus detection by larvae. These modifications standardize experimental conditions, reduce variability, and improve the accuracy of motion tracking in behavioral assays [107,110]. Control groups exposed to stationary patterns serve as internal references to validate motion perception specificity, a methodology also suitable for OKR assessments. Recently, standardized protocols for locomotion-based OMR assays have been proposed, particularly for developmental neurotoxicity screening [111].

### 4.5. Visual Startle Response (VSR) and Visual Motor Response (VMR) Assays 

Although various tests have been developed to assess vision in zebrafish, two of them particularly rely on stereotyped swimming behaviors: the Visual Startle Response (VSR) and the Visual Motor Response (VMR), both of which are suitable for high-throughput analysis. The VSR refers to a rapid, involuntary reaction to an unexpected sensory stimulus, often used as an indirect measure of visual perception. The ability to respond quickly to visual stimuli (e.g., a sudden change in light) is, indeed, linked to retinal function. In zebrafish, the visual startle response becomes detectable as early as 68 hpf, coinciding with the initial formation of functional synapses in the outer and inner plexiform layers of the retina [112]. In a typical VSR assay, zebrafish larvae—usually at 5–7 dpf—are placed individually in a multi-well plate on a microscope stage. A shutter mechanism over a light source is used to generate rapid light transitions [113]. To ensure consistency, larvae are usually habituated in environmentally favorable light conditions for a set period before testing. The assay may include multiple rounds of light transitions to assess response reliability. However, at this stage, larvae also begin displaying spontaneous swimming movements, making the startle response less suitable for genetic screening. Additionally, this assay only evaluates the ability to distinguish light from dark, without providing insights into the development of form vision [67]. The Visual Motor Response (VMR) measures both the initial spike in locomotion (startle response) following light transitions and the time required to return to the baseline activity [113]. It can be detected in larvae as early as 3 dpf, and becomes robust by 5 dpf [113,114]. VMR assays typically involve acclimating 5 dpf larvae for 1–3 h before subjecting them to multiple rounds of on-off light transitions, each lasting 30 min. In a standard protocol of VMR, each larva is placed in an individual well within a multi-well plate, which is then enclosed in a lightproof chamber to minimize external disturbances and ambient light interference, but other formats have also been used [115,116]. Regardless of the system used, the recorded VMR typically consists of two distinct phases: a response to sudden light onset (Light-on VMR) and a response to sudden light offset (Light-off VMR). The movements of larvae are tracked in real-time using infrared illumination and recorded by an infrared camera inside the system [113,117,118]. A retinal-degeneration zebrafish model, *pde6cw59*, showed significantly reduced VMR compared to the WT larvae; however, the treatment with the naturally derived compound Schisandrin B ameliorates the light sensation and reduces the size of the abnormally large rods [119]. Both the VSR and VMR assays are valuable tools for high-throughput drug screening in retinal degenerative diseases, offering a means to identify compounds that enhance light perception and potentially restore visual function.

### 4.6. The Light/Dark Preference Test in Zebrafish: Insights into Visual Processing and Neural Adaptation

The light-transition test is a behavioral assay designed to assess visual function, photoreceptor activity, and neurological responses to changes in light conditions. This test leverages the characteristic swimming patterns of zebrafish larvae in response to alternating light and dark phases, a behavior that emerges following swim bladder inflation at 4 dpf [120]. Typically, zebrafish exhibit low basal locomotor activity under light conditions, followed by a sharp increase in movements when the light is switched off, a response that reflects phototactic behavior [121]. The assay is commonly performed using automated video-tracking systems, which provide quantitative measures of locomotor activity and response dynamics. These analyses enable the detection of subtle behavioral alterations indicative of neurological or visual dysfunction. However, variations in experimental conditions—including differences in light intensity, genetic background, and developmental stage—can influence test outcomes and must be carefully controlled. A published systematic review highlights significant protocol discrepancies, particularly regarding zebrafish age, type and size of well plates used, and the duration of light and dark phases [120,122]. To account for these variations, Hillman et al. proposed a standardized protocol using zebrafish larvae aged from 96 to 120 hpf [123]. In this protocol, larvae are placed in 48-well plates filled with a defined medium (E3 medium) to ensure a stable aquatic environment. The assay is conducted in a chamber with controlled LED lighting, habituating larvae to the environment for 30 min. This critical step should enhance locomotor responses in the dark phase, improving data reliability. Following habituation, larvae are subjected to alternating light (e.g., 500 lux) and dark (0 lux) phases. Shorter light-dark cycles (e.g., 5 min) are effective for evaluating immediate responses and rapid habituation, while longer cycles (e.g., 15 min) allow for assessments of sustained locomotor activity and prolonged stress responses. A 5-min alternation over 30 min has been identified as the optimal protocol for maintaining consistent locomotor activity while minimizing habituation effects.

### 4.7. Habituation to Repeated Stimuli: A Model for Sensory Adaptation and Neural Plasticity

An extension of the light-transition assay is the habituation to repeated stimuli test, which assesses the ability of zebrafish larvae to filter out repetitive, non-threatening stimuli. Habituation represents the simplest form of non-associative learning, where repeated exposure to a stimulus leads to a progressive reduction in response intensity [124,125]. This phenomenon reflects adaptive neural processes that allow organisms to focus on novel or biologically relevant stimuli while ignoring predictable environmental cues. To investigate habituation mechanisms, zebrafish larvae are exposed to repeated light-dark transitions or looming stimuli at regular intervals (e.g., every 30 s). Looming stimuli mimic an approaching object—potentially a predator—engaging neurons in the optic tectum and pretectum responsible for processing expanding edges and triggering escape behaviors. This response relies on specialized neurons encoding angular expansion, signaling imminent collision, and initiating survival-critical maneuvers [126]. Understanding how zebrafish detect, process, and habituate to these visual cues provides crucial insights into the neural dynamics underlying survival behaviors. The retina plays a central role in detecting changes in light intensity, color, and motion. For looming stimuli, retinal photoreceptors capture the expanding edges of an approaching object, transmitting information to specialized looming-sensitive (LS) neurons in the optic tectum. These neurons, activated by expanding visual stimuli, signal potential threats. In parallel, dimming-sensitive (DS) neurons modulate escape responses by inhibiting LS neuron activity through GABAergic modulation. DS neurons project inhibitory signals from the optic tectum to LS neurons in the pretectum, with further integration occurring in the habenula. The habenula then relays inhibitory signals to downstream structures, such as the interpeduncular nucleus (IPN), suppressing excessive escape responses [127]. Experimental approaches for investigating habituation typically include full-field dimming, expanding edges, or combined stimuli, allowing researchers to isolate specific visual processing pathways. Behavioral responses are recorded over 20–30 trials using high-speed cameras, ensuring precise tracking of movement parameters such as tail flick amplitude, latency, and directional accuracy. Adequate recovery periods between trials help differentiate habituation from sensory adaptation, as true habituation involves synaptic plasticity rather than peripheral receptor desensitization [128]. Quantifying habituation can reveal neural adaptations at multiple levels. For example, reduced activity in LS neurons and potentiation of DS neuron responses during repeated stimulation can be visualized using two-photon calcium imaging [126,129]. This technique enables real-time monitoring of neuronal dynamics, providing a direct correlation between behavioral habituation and underlying neural activity. Pharmacological manipulations further elucidate the molecular basis of habituation. Long-term habituation depends on N-Methyl-D-aspartate (NMDA) receptor-mediated synaptic plasticity, a fundamental mechanism for memory formation. NMDA receptor antagonists block long-term habituation, impairing the suppression of escape responses to non-threatening stimuli and highlighting the role of glutamatergic signaling in adaptive learning [130]. Similarly, disruption of GABAergic pathways leads to exaggerated responses, as inhibition of DS neurons fails to regulate LS-mediated escape behaviors [125,126]. Habituation is a critical process that allows animals to conserve energy by filtering irrelevant stimuli. Disruptions in this balance—such as impairments in NMDA receptor signaling, GABAergic inhibition, or habenula function—can lead to heightened sensitivity to sensory inputs, mimicking features of neurodevelopmental disorders, including autism spectrum disorder (ASD) and attention deficit hyperactivity disorder (ADHD). Studying these neural circuits in zebrafish provides valuable insights into the mechanisms underlying sensory processing abnormalities and offers a potential platform for therapeutic screening. By refining behavioral assays such as the light-transition test and habituation paradigms, researchers can develop standardized protocols to investigate neural plasticity, sensory adaptation, and the role of neurotransmitter systems in modulating escape behaviors. These studies contribute to a broader understanding of neurobiological mechanisms in both health and disease, paving the way for targeted interventions in neurological disorders characterized by sensory dysfunction.

### 4.8. The Phototactic Behavior Assay: Evaluating Visual Function and Light-Driven Responses

The phototactic behavior assay investigates the innate tendency of zebrafish larvae to move toward illuminated areas (positive phototaxis), a behavior linked to survival mechanisms such as safety and foraging [131]. Zebrafish larvae possess UV-sensitive cones (SWS1 opsins), which develop by 72 hpf and work as both a protective mechanism against photodamage and an essential component of prey capture. This behavior is mediated by melanopsin-expressing retinal ganglion cells (RGCs) and other opsin-based systems involved in light detection and processing [132,133]. The experimental setup typically consists of a dual-chamber apparatus with a brightly illuminated compartment, and another kept dark or under UV light (~360 nm). Habituation is critical for ensuring reliability, as sudden brightness changes can overwhelm visual processing, potentially altering behavioral outcomes [134]. Furthermore, the test can be carried out with blue, green, and red lights in the lighting room to assess the effect on color perception. To determine whether the larvae entered the chamber unintentionally, a controlled experiment should be conducted in which the chamber remains dark after the partition is lifted [135]. During the assay, groups of 10–15 zebrafish larvae (72–120 hpf) are placed in a neutral central zone, and their movement is tracked using automated systems such as EthoVision XT by Noldus. However, alternative custom behavior recording systems have also been successfully implemented, combining infrared illumination, high-resolution cameras, and programmable LED arrays to finely control stimulus presentation and capture locomotor responses [136]. These setups, while technically demanding, offer a cost-effective and highly detailed behavioral profiling option compared to commercial tracking tools. The time spent in the illuminated, dark, or UV-exposed chambers is then quantified to identify phototactic behavioral patterns and potential associations with anxiety-like responses or retinal dysfunctions. This assay serves as a powerful tool for evaluating the functional integrity of the visual system and for exploring how external factors—such as pharmacological treatments, environmental toxins, or genetic modifications—alter phototactic behavior and light sensitivity. In addition to its applications in visual neuroscience, it is widely used in environmental toxicology; for instance, exposure to UV stabilizers like benzotriazoles has been shown to disrupt phototactic responses, suggesting potential neurotoxic effects [137]. The assay is also suitable for screening anxiolytic compounds by assessing dark-avoidance behavior in zebrafish larvae [131]. Through phototactic response analysis, researchers can investigate visual processing, behavioral adaptations to light stimuli, and the broader effects of environmental or pharmacological interventions—making this assay a versatile approach for both basic neuroscience and translational research.

### 4.9. Color Perception Assay: Investigating Wavelength Sensitivity and Visual Processing

The color perception assay evaluates zebrafish larvae’s sensitivity to different wavelengths of light, providing valuable insights into the development and function of cone photoreceptors involved in visually guided behaviors. Zebrafish possess a tetrachromatic retina with cones tuned to UV (~360 nm), blue (~415 nm), green (~480 nm), and red (~570 nm) wavelengths, allowing them to detect a broad spectrum of colors crucial for ecologically relevant behaviors such as prey detection and predator avoidance [137]. To assess color perception, experiments typically employ LED-based visual systems calibrated to match the spectral sensitivities of zebrafish cones. Larvae are placed in dual-choice chambers or multi-color arenas, and their movements are recorded using automated tracking tools like EthoVision, which enable high-throughput and precise behavioral quantification. However, while these systems offer significant advantages, alternative approaches have also been successfully developed. Notably, some studies have adopted custom-built behavioral setups, integrating infrared illumination, high-resolution CMOS cameras, and programmable LED arrays to precisely deliver visual stimuli and record locomotor responses across a broad range of wavelengths [136,138]. Although more technically complex, these tailored systems allow for fine-grained behavioral analysis and represent a cost-effective and flexible alternative to commercial solutions. These approaches, while less commercially standardized, provide robust and reproducible results when paired with careful experimental design and appropriate controls. The time larvae spend in different color zones offers quantitative measures of their color preference and discrimination ability [139]. Zebrafish larvae typically show a strong phototactic response to blue light, which enhances prey visibility in aquatic environments where shorter wavelengths penetrate more deeply [140]. Conversely, they tend to avoid UV light, likely as a protective mechanism during early developmental stages when their transparency increases vulnerability to photodamage [141]. UV-sensitive circuits are active early on to support prey detection, while sensitivity to longer wavelengths becomes increasingly important as larvae mature and adapt to their visual environment. Color perception assays are also useful for detecting neurotoxic effects on the visual system. For instance, exposure to methylmercury has been shown to damage retinal cells, leading to impaired wavelength discrimination and altered color-driven behaviors [54]. Moreover, multi-color paradigms allow researchers to explore how zebrafish resolve competing visual cues, offering insights into higher-order visual processing and decision-making. Overall, zebrafish color vision assays provide a powerful platform for investigating cone function, neural circuit development, and the ecological significance of color-guided behaviors—revealing how vision adapts to environmental challenges and is influenced by internal or external factors.

## 5. Discussion

Early diagnosis of neurodegenerative diseases is essential for effective management, as timely interventions can slow progression and improve patient outcomes. In this context, the retina is emerging as a sensitive and reliable “window” into central nervous system (CNS) pathology, reflecting the strong correlation between retinal integrity and brain health [142,143]. Numerous studies have demonstrated associations between retinal thickness, function, and disease severity across various neurodegenerative conditions [49,144,145,146]. Thanks to its shared embryological origin, anatomical parallels, and physiological pathways with the brain, the retina offers a unique, non-invasive model for detecting CNS dysfunction. Retinal alterations—such as oxidative stress, neuroinflammation, mitochondrial deficits, and synaptic loss—often represent early markers of neurodegeneration, making the retina an ideal system to investigate neuronal and microvascular damage [143]. Unlike the brain, which is protected by the blood–brain barrier, the retina is readily accessible, enabling in vivo imaging and pharmacological testing. This accessibility enhances its value for studying disease mechanisms and for screening therapeutic interventions [147]. Notably, therapeutic effects observed in the retina often mirror those in the brain, reinforcing its relevance in drug discovery [147]. Zebrafish have become a valuable model in neurodegenerative research due to high genetic and protein homology with humans, rapid development, and optical transparency, which facilitate detailed analysis of the nervous system [148,149]. Their retinal structure, adapted for daylight vision and characterized by a high cone-to-rod ratio, differs significantly from that of nocturnal rodents such as mice and rats, and makes zebrafish particularly suitable for studying retinal disorders [39,40,150]. From an ethical perspective, zebrafish comply with the 3Rs principle, reducing reliance on higher vertebrates. Their small size, high fecundity, and transparent larvae enable non-invasive imaging and behavioral testing [151]. Larvae also display robust, quantifiable visual behaviors, enhancing the model’s value in vision research. The use of behavioral assays in zebrafish larvae offers the potential to reduce the need for mammalian models while maintaining high scientific rigor [152,153,154,155]. In contrast, behavioral studies in adult zebrafish are less common due to greater susceptibility to confounding factors, including individual variability, learning effects, environmental influence, motor artifacts, and the need for invasive procedures. It is worth noting that adult zebrafish are rarely used in behavioral studies of retinal dysfunction, mainly due to several confounding factors, including individual variability, environmental influences, learning and habituation effects, observer and experimenter bias, as well as motor and non-visual factors. Additionally, the use of invasive procedures further limits their applicability. For these reasons, this review focuses on key behavioral assays developed to assess visual function during the early developmental stages of the zebrafish life cycle—from basic light detection to complex visuomotor coordination (Table 2). These assays provide rapid and sensitive readouts of retinal health and visual processing, often identifying functional impairments before structural damage becomes apparent. By organizing behavioral tests according to developmental milestones, researchers can systematically explore the maturation of visual function in zebrafish, allowing for precise evaluations of both normal development and pathological conditions. These assays are especially effective in detecting early functional deficits indicative of neurodegenerative or neurodevelopmental disorders and are increasingly employed to assess the efficacy of potential therapeutic interventions. However, methodological heterogeneity across laboratories remains a major challenge, as variations in experimental protocols can significantly impact the reliability and reproducibility of findings. Critical factors such as the developmental stage, stimulus calibration, and the integration of behavioral, electrophysiological, and imaging data must be carefully standardized to ensure consistency and comparability across studies. Moreover, potential confounding variables—such as locomotor activity, stress, or cognitive load—may bias behavioral outcomes. Therefore, a multimodal approach combining behavioral assays with electrophysiological recordings and high-resolution imaging will yield a more comprehensive understanding of retinal function and visual processing.

## 6. Conclusions

The integration of advanced retinal imaging, behavioral testing, and zebrafish models constitutes a powerful strategy for the early detection, mechanistic exploration, and therapeutic evaluation of neurodegenerative diseases. To fully harness the potential of this approach, future research should prioritize the standardization of experimental protocols and the integration of multimodal datasets. This will facilitate the discovery of novel biomarkers and innovative treatments. By combining these methodologies, researchers can accelerate drug development, enhance translational research, and ultimately contribute to improved clinical outcomes for patients affected by neurodegenerative conditions.

## Figures and Tables

**Figure 1 biology-14-00934-f001:**
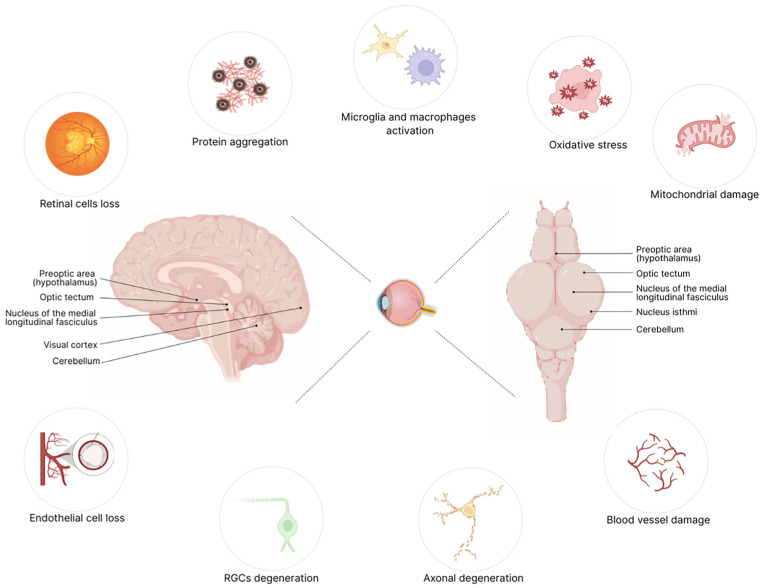
**The retina is a window to brain degeneration.** Shared degenerative mechanisms between the retina and the brain highlight their common developmental and functional origin as components of the CNS. This illustration points to converging pathological features—oxidative stress, protein aggregation, mitochondrial damage, axonal degeneration, microglial activation, and vascular dysfunction—observed in both retinal and cerebral neurodegeneration. Given the retina’s accessibility, it is a valuable model for investigating early neurodegenerative events, offering critical insights into disease mechanisms. The image also depicts structural similarities between the human brain and the zebrafish brain, further supporting the use of zebrafish as a translational model in neurodegenerative research.

**Figure 2 biology-14-00934-f002:**
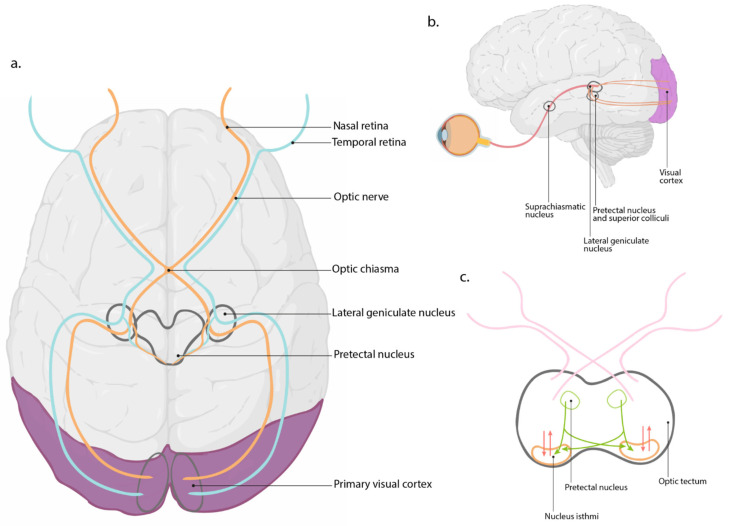
**Visual pathway from retina to brain**. (**a**) Superior view of the visual circuit in the human brain. Signals from the nasal retina cross at the optic chiasm, while those from the temporal retina remain ipsilateral. These projections synapse at the lateral geniculate nucleus and then relay to the primary visual cortex. Additional projections reach the pretectal nucleus. (**b**) Lateral view of the visual circuit in the human brain. (**c**) Schematic zebrafish visual circuit: The optic tectum is the principal visual processing center in zebrafish, mediating sensorimotor information for visually guided behaviors. The pretectal nucleus, which receives input from the tectum, sends inhibitory projections to the nucleus isthmi, modulating visuomotor reflexes like the optokinetic response. The nucleus isthmi forms reciprocal connections with the optic tectum and contributes to visual attention and stimulus selection.

**Figure 3 biology-14-00934-f003:**
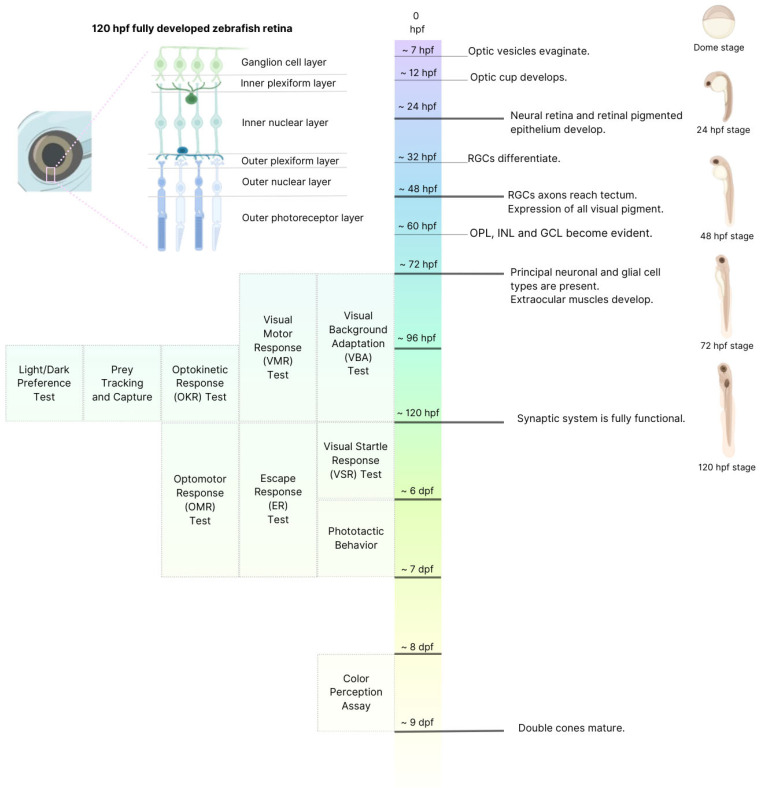
**Development of visual circuits and related behavior assays in larval zebrafish.** Schematic timeline of visual network development in zebrafish with functional behavioral assays available at specific timepoints to elicit precise visual behavioral responses. On the top left side is a close-up schematic view of retinal structure and cellular organization in zebrafish retina, highlighting the layered organization fully developed at 5 days post fertilization (dpf). Retinal layers include, as in humans, the ganglion cell layer, inner and outer plexiform layers (synaptic layers), inner and outer nuclear layers, and the outer photoreceptor layer, which contains rods and cones. Each layer is pivotal in processing visual information from photoreception to signal transmission to the brain.

**Table 1 biology-14-00934-t001:** Main human neurodegenerative diseases associated with retinal impairments.

Disease	Retinal Alterations/Visual Dysfunction	References
**Alzheimer’s disease (AD)**	Retinal nerve fiber layer (RNFL) thinning, reduced blood perfusion, macular vessel density and inner retina thinning.	[4,5,6,7,8,9]
**Parkinson’s Disease (PD)**	RNFL thinning, optic neuritis, microcystic macular edema, cataracts, vision loss, photopic contrast sensitivity.	[10,11,12,13,14,15]
**Multiple Sclerosis** **(MS)**	RNFL thinning, optic neuritis, microcystic macular edema, decreased visual acuity, and nystagmus.	[16,17,18]
**Huntington’s Disease (HD)**	RNFL thinning, retinal ganglion cell loss, impaired visual processing.	[19,20,21]
**Charcot-Marie-Tooth Disease (CMT)**	Optic atrophy.	[22]
**Mitochondrial Disorders**	Ocular involvement, vision loss, ocular motility alterations.	[23]
**Frontotemporal dementia**	RNFL thinning, difficulty with eye movements, changes in visual perception.	[24,25,26]
**Mild Cognitive impairment (MCI)**	Reduced peripapillary RNFL and macular thickness, retinal microvascular density loss, correlated with cognitive impairment severity.	[5,27]
**Hereditary Ataxias: Friedreich’s ataxia (FRDA) and spastic ataxia of Charlevoix-Saguenay (ARSACS)**	Retinal ganglion cell loss and abnormal visual evoked potentials.	[5,28,29,30]
**Spinocerebellar Ataxias (SCA)**	Retinal ganglion cell loss, abnormal visual evoked potentials, progressive visual loss, macular degeneration, and optic atrophy.	[30,31,32,33]
**Hereditary Spastic Paraparesis (HSP)**	Pigmentary retinal degeneration, ophthalmoplegia, optic atrophy, cataracts, and nystagmus.	[34]
**Lafora Disease**	Rods and cones dysfunction, progressive visual impairment.	[35]

**Table 2 biology-14-00934-t002:** Summary of each considered visual behavior test with the respective developmental stage (from 3 to 9 dpf), purpose, method, and visual function assessed.

Test	Developmental Stage	Purpose	Method	Visual Function Assessed
**Visual Background Adaptation (VBA)**	from–3 dpf	Assesses neuroendocrine response to ambient light	In bright light, melanophores contract; in darkness, melanin disperses	Retinal ganglion cell function, light adaptation
**Visual Motor Response (VMR)**	3–5 dpf	Assesses time to return to base activity after sudden visual stimuli	Larvae are exposed to multiple rounds of on-off light transitions, each lasting 30 min	Motion detection and visual motor coordination
**Prey Tracking and Capture**	from4 dpf	Tests visual tracking and object recognition	Larvae are introduced to moving prey (e.g., rotifers), and capture success is observed	Visual acuity, contrast sensitivity, motion tracking
**Escape response (ER**)	5–7 dpf	Evaluates visual threat detection and sensorimotor integration	Zebrafish are exposed to a looming stimulus (e.g., black stripe in a rotating drum, virtual predator, or robotic model), and their escape behavior (latency, trajectory, response probability) is analyzed	Motion perception, threat avoidance, visual processing, and circadian modulation of visual sensitivity
**Optokinetic Response (OKR)**	from 4 to 5 dpf	Measures visual motion processing	Zebrafish are exposed to rotating striped patterns, and eye movements (nystagmus) are recorded	Retinal function, motion detection, visual acuity
**Optomotor Response (OMR)**	5–7 dpf	Evaluates motion detection and visuomotor coordination	Zebrafish swim in response to moving patterns (e.g., drifting gratings)	Motion perception, contrast sensitivity, visual-motor integration
**Visual Startle Response (VSR)**	5–6 dpf	Tests reaction to sudden visual stimuli	Sudden flashes or moving objects elicit a startle reflex	Motion detection, contrast sensitivity, visual-motor coordination
**Light/Dark Preference Test**	4–5 dpf	Assesses innate preference for light or dark environments	Zebrafish are placed in a tank with light and dark zones, and their position is recorded	Light perception, photoreceptor function, scotopic/photopic vision
**Phototactic Behavior**	6–7 dpf	Measures attraction or aversion to light stimuli	Zebrafish are given a choice between illuminated and dark areas	Light sensitivity, retinal function, visual-driven locomotion
**Color Perception Assay**	8–9 dpf	Evaluates the ability to discriminate colors	Zebrafish are trained to associate colored stimuli with a reward, or tested for innate color preference	Cone photoreceptor function, color discrimination, photopic vision

## Data Availability

Not applicable. No new data were generated or analyzed in this review paper.

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
