# Peer review of "Eyes Wide Open: Assessing Early Visual Behavior in Zebrafish Larvae"

_biology, 2025, doi:10.3390/biology14080934_

Round 1
Reviewer 1 Report
Comments and Suggestions for Authors
Overview:
The review presents and discusses the main behavioral assays using the zebrafish experimental model. It analyzes how early visual behavior in zebrafish larvae can serve as a tool for the early detection of neurodegenerative diseases, considering that the retina shares both origin and functions with the brain. This provides a non-invasive and efficient strategy for early diagnosis and high-throughput drug screening.
An excellent review, congrats.
Suggestions:
-
Presentation figure: I suggest revising the introductory figure to ensure it matches the quality of the others.
-
Regarding the behavioral tests: as a suggestion, first describe the test, and then proceed to discuss it, highlighting its advantages and disadvantages
Author Response
The review presents and discusses the main behavioral assays using the zebrafish experimental model. It analyzes how early visual behavior in zebrafish larvae can serve as a tool for the early detection of neurodegenerative diseases, considering that the retina shares both origin and functions with the brain. This provides a non-invasive and efficient strategy for early diagnosis and high-throughput drug screening.
An excellent review, congrats.
- Presentation figure: I suggest revising the introductory figure to ensure it matches the quality of the others.
We thank the reviewer for the suggestion. In response, we have revised the graphical abstract figure to improve its visual quality and overall coherence with the other figures in the manuscript. The updated version ensures a more professional and consistent presentation throughout the paper.
- Regarding the behavioral tests: as a suggestion, first describe the test, and then proceed to discuss it, highlighting its advantages and disadvantages
We thank the reviewer for the helpful suggestion. We have restructured the sections describing behavioral tests to first present each assay clearly—focusing on its experimental design and procedures—before discussing its applications, advantages, and limitations. This revised organization improves the logical flow and enhances the clarity of each subsection. The modifications are highlighted in the revised manuscript, as suggested, to facilitate the review of the updated structure and content.
Reviewer 2 Report
Comments and Suggestions for Authors
biology-3702843-peer-review-v1
Eyes wide open: assessing early visual behavior in zebrafish larvae
Michela Giacich, Maria Marchese, Devid Damiani, Filippo Maria Santorelli, and Valentina Naef
This review explores the zebrafish (Danio rerio) as a model for studying retinal degeneration and its connection to neurodegenerative diseases. Highlighting the structural and functional similarities between the retina and the brain, the article discusses behavioral assays.
I would like to suggest two inclusions that can enrich the article, providing visual and organized information that complements the text.
- Table on neurodegenerative diseases with visual impairment: A table listing neurodegenerative diseases associated with structural and functional retinal degeneration could help readers better understand the connections between these conditions and visual impacts. It could also highlight the relevance of the retina as a biomarker for early diagnosis.
Figure of the map of brain regions interconnected with the retina: A figure illustrating the connections between brain regions and retinal neuronal structures could facilitate the visualization of the functional and anatomical interactions described in the text. This would be especially useful for readers who are not familiar with neuroanatomy.
These additions could improve the clarity and understanding of the article, making it more informative and visually appealing.
Author Response
This review explores the zebrafish (Danio rerio) as a model for studying retinal degeneration and its connection to neurodegenerative diseases. Highlighting the structural and functional similarities between the retina and the brain, the article discusses behavioral assays.
I would like to suggest two inclusions that can enrich the article, providing visual and organized information that complements the text.
- Table on neurodegenerative diseases with visual impairment: A table listing neurodegenerative diseases associated with structural and functional retinal degeneration could help readers better understand the connections between these conditions and visual impacts. It could also highlight the relevance of the retina as a biomarker for early diagnosis.
We thank the Reviewer for this valuable suggestion. Following the recommendation, we have included a new table (now Table 1) that summarizes the main human neurodegenerative diseases associated with retinal alterations—both structural and functional. This addition is intended to provide a clear and concise overview of the relationship between each condition and the type of retinal involvement, supporting the concept of the retina as a potential non-invasive biomarker for early diagnosis and disease monitoring. We believe this table improves the clarity and informative value of the manuscript. The modifications are highlighted in the revised manuscript, as suggested, to facilitate the review of the updated structure and content.
- Figure of the map of brain regions interconnected with the retina: A figure illustrating the connections between brain regions and retinal neuronal structures could facilitate the visualization of the functional and anatomical interactions described in the text. This would be especially useful for readers who are not familiar with neuroanatomy.
We thank the Reviewer for this insightful suggestion. In response, we have added a new figure (Figure 2) that schematically represents the anatomical and functional connections between the retina and key brain regions involved in visual processing and neurodegeneration. The modifications are highlighted in the revised manuscript, as suggested, to facilitate the review of the updated structure and content. These additions could improve the clarity and understanding of the article, making it more informative and visually appealing.
Reviewer 3 Report
Comments and Suggestions for Authors
Robust experimental model: In biomedical research, the use of zebrafish as an experimental animal model is important, as it represents a viable alternative in general, and particularly for eye research. This concept is clearly conveyed by the authors in this review.
Appropriate technical language:The manuscript employs scientific language that is both accessible and specialized.
Structure and sequence of the review: The text is clearly written and logically structured: it first frames the problem (early diagnosis), then introduces the model (zebrafish), describes the methods (behavioral assays), defines their advantages, and finally highlights a current challenge (standardization).
Relevance of the topic: The link between the retina and neurodegeneration is well justified as a promising area of study. Moreover, the positioning of the zebrafish as a functional model for this field is both timely and appropriate.
Comprehensive coverage:The review addresses both methodological aspects (behavioral assays) and practical applications (pharmacological screening, early diagnosis), as well as current challenges in the field (protocol variability).
Multidimensional analysis: The article includes well-targeted techniques for studying the visual pathway -from information input, transduction, and central integration to behavioral analysis- focusing on critical stages of individual development.
Techniques and methodological data: In particular, Table 1 serves as an excellent timeline reference for the possible techniques and studies that can be applied in this species to yield robust results. Likewise, the figures are sufficiently descriptive to support understanding of the manuscript.
General relevance:Although all the techniques described have been widely employed in numerous studies, this review stands out for its proposed approach, guiding the selection of techniques based on the research question and the developmental stage of the individual.
Points that should be addressed
Writing and grammar: The manuscript begins by highlighting the features that make zebrafish a key model for studying retinal degeneration. However, this statement is repeated unnecessarily throughout the text, leading to redundancy.
Figures: Although Figure 1 is descriptive, it depicts only human diseases, even though not all of them have been directly studied in humans. It is suggested to also include an image of the zebrafish brain to visually emphasize which diseases have been effectively modeled in this organism.
Additional functional validations: It is recommended to include genetic and behavioral rescue experiments to confirm that the observed effects are specifically related to the pathology under study. Furthermore, from subsection 4.7 onward, the description of techniques and their applications becomes more detailed. It would be advisable to provide the same level of detail in the preceding sections to ensure consistency and depth throughout the document.
Alternative techniques: In subsection 4.9, “Color Perception Assay: investigating wavelength sensitivity and visual processing,” the use of automated systems such as EthoVision is suggested for assessing the sensitivity of zebrafish larvae to different wavelengths of light. However, since not all laboratories have access to such equipment, it would be valuable for the authors to suggest alternative methods that allow these studies to be carried out with more accessible resources.
Comments on the Quality of English Language
It is recommended to revise the overall writing and correct grammatical and spelling errors in English to enhance clarity and improve overall comprehension.
Author Response
Robust experimental model: In biomedical research, the use of zebrafish as an experimental animal model is important, as it represents a viable alternative in general, and particularly for eye research. This concept is clearly conveyed by the authors in this review.
Appropriate technical language:The manuscript employs scientific language that is both accessible and specialized.
Structure and sequence of the review: The text is clearly written and logically structured: it first frames the problem (early diagnosis), then introduces the model (zebrafish), describes the methods (behavioral assays), defines their advantages, and finally highlights a current challenge (standardization).
Relevance of the topic: The link between the retina and neurodegeneration is well justified as a promising area of study. Moreover, the positioning of the zebrafish as a functional model for this field is both timely and appropriate.
Comprehensive coverage:The review addresses both methodological aspects (behavioral assays) and practical applications (pharmacological screening, early diagnosis), as well as current challenges in the field (protocol variability).
Multidimensional analysis: The article includes well-targeted techniques for studying the visual pathway -from information input, transduction, and central integration to behavioral analysis- focusing on critical stages of individual development.
Techniques and methodological data: In particular, Table 1 serves as an excellent timeline reference for the possible techniques and studies that can be applied in this species to yield robust results. Likewise, the figures are sufficiently descriptive to support understanding of the manuscript.
General relevance:Although all the techniques described have been widely employed in numerous studies, this review stands out for its proposed approach, guiding the selection of techniques based on the research question and the developmental stage of the individual.
Points that should be addressed
Writing and grammar: The manuscript begins by highlighting the features that make zebrafish a key model for studying retinal degeneration. However, this statement is repeated unnecessarily throughout the text, leading to redundancy.
We appreciate the reviewer’s observation. We have carefully revised the manuscript to eliminate redundant statements regarding the advantages of the zebrafish model. These features are now clearly stated in the introduction and referenced more concisely in subsequent sections to avoid unnecessary repetition, while still maintaining clarity for the reader.
Figures: Although Figure 1 is descriptive, it depicts only human diseases, even though not all of them have been directly studied in humans. It is suggested to also include an image of the zebrafish brain to visually emphasize which diseases have been effectively modeled in this organism.
We thank the reviewer for this insightful suggestion. In response, we have revised Figure 1 to include an illustration of the zebrafish brain. We believe that this modification improves the clarity and relevance of the figure, aligning it more closely with the scope of the review.
Additional functional validations: It is recommended to include genetic and behavioral rescue experiments to confirm that the observed effects are specifically related to the pathology under study. Furthermore, from subsection 4.7 onward, the description of techniques and their applications becomes more detailed. It would be advisable to provide the same level of detail in the preceding sections to ensure consistency and depth throughout the document.
We sincerely thank the reviewer for their valuable suggestions. Regarding the inclusion of genetic and behavioral rescue experiments, we fully agree that these approaches are crucial to validate the specificity of observed phenotypes. However, the aim of this review is to provide an overview of behavioral assays used to assess visual function in zebrafish larvae, rather than to describe in detail specific disease models and associated rescue strategies. A thorough discussion of genetic manipulations and rescue experiments would exceed the scope of the present work and would be more appropriate for a dedicated review.
We also appreciate the reviewer’s comment on the varying level of detail across different sections. As mentioned in the manuscript, this variability reflects the current state of the literature, where experimental protocols and methodological descriptions differ significantly between studies. Consequently, it is not always possible to provide a uniform level of detail for each assay, as some are better characterized or more extensively reported than others. One of the goals of this review is precisely to underline these inconsistencies and to emphasize the need for standardized, well-described procedures to improve reproducibility and cross-study comparisons.
Alternative techniques: In subsection 4.9, “Color Perception Assay: investigating wavelength sensitivity and visual processing,” the use of automated systems such as EthoVision is suggested for assessing the sensitivity of zebrafish larvae to different wavelengths of light. However, since not all laboratories have access to such equipment, it would be valuable for the authors to suggest alternative methods that allow these studies to be carried out with more accessible resources.
We thank the Reviewer for this important observation. While automated systems such as EthoVision are indeed useful for high-throughput and quantitative tracking, we agree that not all laboratories have access to such equipment. Therefore, we have revised the manuscript to include a brief overview of alternative approaches that are documented in the literature. These include manual video recording with subsequent analysis (e.g., by time spent in colored zones), the use of simple dual- or multi-choice arenas with direct observation or frame-by-frame image analysis, and open-source tracking software such as Fiji/ImageJ or ZebTrack. These methods, while less automated, still provide reliable data on color preference and wavelength sensitivity when combined with careful experimental design and proper controls. The modifications are highlighted in the revised manuscript, as suggested, to facilitate the review of the updated structure and content.
Reviewer 4 Report
Comments and Suggestions for Authors
The review by Michela Giacich et al., entitled “Eyes wide open: assessing early visual behavior in zebrafish larvae”.
This review is of high interest to the scientific community. The manuscript is well-written and well-organized, with clear, easy to read sections and thoroughly documented references.
Major point:
Please include, if possible, a section discussing examples of how zebrafish have contributed to early diagnosis of neurodegenerative diseases, particularly those associated with the retina. Include examples of behavioral tests used for early diagnosis.
Minor point:
Reference 32 and 33 are the same. Reference 32 is missing in the text.
Author Response
The review by Michela Giacich et al., entitled “Eyes wide open: assessing early visual behavior in zebrafish larvae”.
This review is of high interest to the scientific community. The manuscript is well-written and well-organized, with clear, easy to read sections and thoroughly documented references.
Major point:
Please include, if possible, a section discussing examples of how zebrafish have contributed to early diagnosis of neurodegenerative diseases, particularly those associated with the retina. Include examples of behavioral tests used for early diagnosis.
We sincerely thank the reviewer for their valuable suggestion to include examples of how zebrafish have contributed to the early diagnosis of neurodegenerative diseases, particularly those associated with retinal pathology, as well as related behavioral tests. We fully acknowledge the importance of this topic and agree that zebrafish models have significantly advanced our understanding in this area. However, the primary aim of this review is to provide the scientific community with an overview of behavioral assays that can be performed up to 120 hours post-fertilization in zebrafish larvae to assess potential retinal dysfunctions. The review is not intended to describe zebrafish models of neurodegenerative diseases with visual impairments, nor to cover functional assays and pharmacological screenings in such models. Including detailed examples of disease-specific contributions and related behavioral paradigms would extend beyond the intended scope of this manuscript. Such a discussion would be highly interesting and valuable but is better suited for a dedicated, focused review..
Minor point:
Reference 32 and 33 are the same. Reference 32 is missing in the text.
We sincerely thank the reviewer for pointing out this oversight. We have corrected the reference list by removing the duplicate citation (Reference 32 and 33) and ensured that all references are properly cited within the text